# The relationship between hot flashes and fatty acid binding protein 2 in postmenopausal women

Ting-Yu Chen[1]☯, Wan-Yu Huang[2]☯, Ko-Hung Liu[1], Chew-Teng Kor[3], Yi-Chun Chao[1], Hung-Ming Wu[1,4,5]*

1 Inflammation Research & Drug Development Center, Changhua Christian Hospital, Changhua, Taiwan,
2 Department of Pediatrics, Kung-Ten General Hospital, Taichung City, Taiwan, 3 Division of Statistics,
Internal Medicine Research Center, Changhua Christian Hospital, Changhua, Taiwan, 4 Department of
Neurology, Changhua Christian Hospital, Changhua, Taiwan, 5 Graduate Institute of Acupuncture Science,
China Medical University, Taichung, Taiwan

☯ These authors contributed equally to this work.
* 18288@cch.org.tw

pone.0276391

Hospital, Sichuan University, CHINA

**Data Availability Statement:** All relevant data are
within the paper and its Supporting Information
files.

## Abstract

### Introduction

Hot flashes, the most bothering symptom of menopause, are linked to a metabolic inflammation. Due to estrogen deficiency in menopause, dysbiosis is observed. The intestinal barrier affects the interaction of microbiota in healthy or unhealthy individuals. This study investigates the relationship between hot flashes and gut permeability in postmenopausal women.

### Participants and design

In this cross-sectional study, we divided 289 women, aged 40–65 years, into four groups based on their hot-flash severity: $HF_0$: never experienced hot flashes; $HF_m$: mild hot flashes; $HF_M$: moderate hot flashes; $HF_S$: severe hot flashes. The measured variables included the clinical parameters; hot flashes experience; fasting plasma levels of zonulin, fatty acid binding protein 2 (FABP2), endotoxin, and cytokines/chemokines. We used multiple linear regression analysis to evaluate the relationship between hot flashes and the previously mentioned gut barrier proteins.

### Settings

The study was performed in a hospital medical center.

### Results

The hot flashes had a positive tendency toward increased levels of circulating FABP2 (P-trend = 0.001), endotoxin (P-trend = 0.031), high-sensitivity C-reactive protein (hs-CRP) (P-trend = 0.033), tumor necrosis factor alpha (TNF-α) (P-trend = 0.017), and interferon-inducible protein-10 (IP10) (P-trend = 0.021). Spearman's correlation analysis revealed significant correlations of FABP2 with endotoxin, TNF-α, monocyte chemoattractant protein-1,

**Funding:** This study was supported by the grant MOST 108-2314-B-371-004-MY3 from the Ministry of Science and Technology, Taiwan and by the grant 110-CCH-IRP-55 from Changhua Christian Hospital.The funders had no role in study design, data collection and analysis, decision to publish, or preparation of the manuscript.

**Competing interests:** The authors have declared that no competing interests exist.

IP10, and hs-CRP in the 289 postmenopausal women included in this study. Linear regression analysis revealed that hot-flash severity had significant assoiciations with FABP2 ($P$-trend = 0.002), but not with zonulin. After adjusting for body mass index, age, and menopause duration, multivariate linear regression analysis revealed the differences between HFs (% difference (95% confidence interval), 22.36 (8.04, 38.59), $P$ = 0.01) and $HF_0$ groups in terms of FABP2 levels.

## Conclusions

This study shows that hot flashes are significantly associated with FABP2 levels in postmenopausal women. It suggests that severe hot flashes are linked to an increase in intestinal barrier permeability and low-grade systemic inflammation.

## Introduction

Hot flashes are one of the bothersome symptoms in perimenopause and postmenopause [1, 2]. They present as episodic sensations of heat in the facial, nuchal, and chest regions, frequently followed by sweats, palpitation, anxiety, and irritability [3, 4]. Hot flashes may occur during the day or night. They negatively affect the quality of life during menopause. Previous studies found that hot flashes were associated with reduced health-related quality of life outcomes: sleep disorders, anxiety, depression, and cognitive function decline [5, 6].

Increasing evidence shows that hot flashes impact physical health and are associated with impaired lipid profiles [7], endothelial dysfunction [8], imbalanced adipocyte-derived hormones (leptin and adiponectin) [9], and insulin resistance [10] during menopause. Recently, hot flashes have been identified as a novel female-specific cardiovascular risk factor [11]. Huang et al. reported that hot flashes in postmenopausal women are associated with a low-grade systemic inflammation caused by increased levels of interleukin 8 (IL-8) and tumor necrosis factor alpha (TNF-α). Emerging evidence suggests that hot flashes may be closely linked to the development of metabolic and inflammatory disorders. However, the biological mechanisms underlying these associations remain to be further determined.

Human gastrointestinal tract harbors trillions of microorganisms—gut microbiota. The microbiota is involved in a symbiotic relationship with the body, which makes it essential for sex hormones such as estrogen and glucose metabolism [12–14]. Interestingly, estrogen significantly affects the composition and diversity of the intestinal microbiota and the corresponding disease pathways [15]. The evidence shows that estrogen decline and deficiency in menopause alter the gut microbiota, leading to changes in the microbial richness and diversity [16–18]. Several bacterial species independently associated with metabolic risk factors has been identified, suggesting a key environmental role of the microbiome driving metabolic diseases [19] and chronic low-grade inflammation [20].

The intestinal mucosal barrier has physical, anatomical, and immunological elements. The intestinal mucosa is composed of a monolayer of epithelial cells and the lamina propria [21, 22]. Tight junctions are the major compounds involved in joining intestinal epithelia cells and a critical part for proper epithelial barrier functions [23]. Accordingly, the intestinal mucosa is responsible for the absorption of nutrients from the lumen and separation of the potentially toxic luminal content from the host. When this delicate balance at the mucosal interface is disrupted, gut-derived pathogen-associated molecules and toxic factors will leak into the blood circulation, leading to inflammatory and metabolic disorders [20].

Estrogen increases gut epithelial integrity. In menopause, estrogen declines, which may increase epithelial permeability and bacterial translocation and contribute to systemic inflammation and menopause-related metabolic changes (e.g., obesity and metabolic syndrome) [24]. Increased probability of hot flashes is associated with several factors. Sexual hormones (e.g., estrogen) are a primary mediator of hot flashes. Low estrogen concentrations may play a significant role in vasomotor symptoms [25]. However, little is known about how the changes in menopause-associated gut microbiome impact hot flashes. We further hypothesized that hot flashes have a role in gut permeability related to metabolic dysregulation and systemic inflammation in menopausal women. To gain a better understanding of the relationship between hot flashes and gut permeability, a cross-sectional study was conducted in age-matched groups of postmenopausal women with and without hot flashes.

## Subjects and methods

### Participants and study designs

This cross-sectional study includes women aged 40 to 65 years who presented to the Changhua Christian Hospital for health management reasons. Postmenopause was defined as the time after which women had experienced at least 12 consecutive months of amenorrhea. The present study included postmenopausal women who had never experienced hot flashes during the menopause stages or had experienced hot flashes within the three months prior to the study entry. The range of body mass index (BMI) was more than 18 $kg/m^2$. Women were excluded if they were premenopausal and perimenopausal; received hormone replacement therapy; received a medication for chronic systemic diseases, including hyperlipidemia, diabetes, and hypertension; had a BMI $<$ 18 $kg/m^2$; were smoking. Written informed consent was obtained from all participants. This study was approved by the Changhua Christian Hospital Institutional Review Board (ID: CCH IRB No. 210207). The records and information of the participants were anonymized and deidentified prior to data collection and statistical analysis.

### Anthropometric measures

Blood samples were obtained from each participant in the morning after fasting overnight. The samples were centrifuged at 2500 rpm for ten minutes. Then, the plasma specimen was obtained, aliquoted, and stored at $-80\degree C$ without thawing until assay. Wearing light clothing without shoes the participants were measured for their height and weight. BMI was calculated as weight (kg)/height $(m)^2$.

### Hot flashes

In this study, we categorized hot flashes according to the severity that participants reported as previously described [26]. Briefly, we categorized heat sensation without sweating as mild-degree hot flashes; heat sensation followed by sweating and not interfering with daily activities as moderate-degree hot flashes; heat sensation followed by sweating and causing cessation of activity or interruption of sleep, as severe-degree hot flashes. All participants were required to provide information related to menopause and profiles of vasomotor symptoms. Participants were then divided into four groups according to the severity of hot flashes: $HF_0$ group included postmenopausal women who never experienced hot flashes or night sweats; $HF_m$, women who experienced only mild hot flashes and no night sweats; $HF_M$, women who experienced moderate hot flashes but no night sweats; $HF_S$, women who experienced severe hot flashes or night sweats or both at least four days per week.

## Measurements of plasma cytokines and chemokines

We measured the plasma levels of TNF-α, monocyte chemoattractant protein-1 (MCP-1) (also known as CCL2), and interferon-inducible protein-10 (IP10) (also known as CXCL10) using a Millipore Cytokine Three-Plex Panel Assay (MILLIPLEX MAP Human Cytokine/Chemokine Magnetic Bead Panel) (MILLIPLEX MAP kits, EMD Millipore, Billerica, MA, USA). All analyses were performed in accordance with the manufacturer's protocol. The data were read using a Luminex 200 system (Luminex, Austin, TX, USA). Data on cytokines and chemokines were collected and analyzed using an instrument equipped with MILLIPLEX Analyst software (EMD Millipore). The intra- and interassay laboratory coefficients of variation were less than 8% and 10%, respectively.

## Measurements of zonulin and fatty acid binding protein 2

We measured the plasma levels of zonulin and fatty acid binding protein 2 (FABP2) using commercially available enzyme-linked immunosorbent assay kits for zonulin (Wuhan Fine Biotech, Wuhan, China, No. EH1057) and FABP2 (Wuhan Fine Biotech, Wuhan, China, No. EH3251). The samples were diluted 1000 times using a dilution buffer for zonulin assays and were undiluted for FABP2 assays with the competitive binding technique. We added a biotiny-lated tracer to the samples: The intensity of the color was inversely proportional to zonulin and FABP2 concentrations. Samples were then read at 450 nm, and the 6-parameter algorithm was used to form the standard curve and calculate the data. All sample tests were conducted in duplicate. The intra- and inter-assay coefficient variances of the kit were less than 8% and 10%, respectively, for zonulin and FABP2.

## Measurements of circulating endotoxin levels

After diluting the plasma samples 10 times using a dilution buffer, we measured plasma endo-toxin levels using Chromogenic Limulus Amebocyte Lysate Assay (QCL-1000™; Lonza, Walk-ersville, MD, USA). All assays were performed according to the manufacturer's instructions. To prevent endotoxin contamination, endotoxin tests were performed in an endotoxin free facility by using endotoxin-free materials and reagents, including nonpyrogenic plasticware, depyrogenated glassware, and high-purity water.

## Measurements of sexual hormones, high-sensitivity C-reactive protein, and others

The concentrations of follicle-stimulating hormone (FSH) and estradiol were measured according to the standard procedures of the Department of Laboratory Medicine, Changhua Christian Hospital. Briefly, we measured FSH and estradiol in plasma specimens using the Access hFSH Assay and the Access Estradiol Assay, respectively, on the Beckman Access Immunoassay System (Beckman Coulter, Fullerton, CA, USA). The unit measurement of FSH and estradiol was mIU/mL and pg/ml, respectively. For estradiol, the inter- and intra-assay laboratory coefficients of variation were less than 8% and 8.1%, respectively, and for FSH, they were less than 8% and 6%, respectively. We measured the levels of high-sensitivity C-reactive protein (hs-CRP) and total cholesterol (including low-density lipoprotein (LDL) cholesterol, high-density lipoprotein (HDL) cholesterol, and triglycerides (TG)) using an automatic lab instrument at the Department of Laboratory Medicine, Changhua Christian Hospital.

## Statistical analysis

Results are presented as median (interquartile range (IQR)). Kolmogorov–Smirnov test was used to examine whether variables were normally distribution. One-way analysis of variance

(ANOVA) test or Kruskal–Wallis test was used to determine differences between the study groups. Tukey's post hoc tests and Dunn's multiple comparison tests were then conducted to find the significant differences between the groups. The correlations were determined using Spearman's rank correlation test. The association between the variables and the hot flashes status was determined using multivariate linear regression analysis. The percentage difference in each variable was calculated using the formula $100^*(\exp(\beta)-1)$ and 95% CI for interpreting coefficients in the multivariate linear regression model. Statistical analyses were performed using SPSS software version 19.0.0 (IBM Corporation, Somers, NY, USA). Two-tailed $P < 0.05$ was considered statistically significant.

## Results

Two hundred and eighty-nine women fulfilled the inclusion criteria. Enrolled women were divided into four groups based on the severity of hot flashes. Group $HF_0$ included 117 postmenopausal women; $HF_m$, 33; $HF_M$, 41; $HF_S$, 98. No significant differences existed between the groups in terms of the median age, menopausal period, FSH levels, BMI, or lipid profiles, including total cholesterol, HDL, and LDL (Table 1).

### Increased levels of FABP2 and TNF-α in postmenopausal women with hot flashes

The intensity of hot flashes was positively associated with increased levels of circulating FABP2 ($P$-trend = 0.001), endotoxin ($P$-trend = 0.031), hs-CRP ($P$-trend = 0.033), TNF-α ($P$-trend = 0.016), and IP10 ($P$-trend = 0.021). Simultaneously, Kruskal–Wallis test revealed a significant difference existed between the four groups in terms of plasma levels of FABP2 ($P = 0.005$), TNF-α ($P = 0.02$,) and hs-CRP ($P = 0.018$), but not zonulin, (Table 1). Dunn's multiple comparison tests further revealed that women in group $HF_S$ had significantly higher levels of FABP2 and TNF-α than those in group $HF_0$ ($P < 0.05$). No significant differences existed in terms of the values of these two parameters between groups HF0, $HF_m$, and $HF_M$.

### FABP2 and zonulin levels are mainly associated with inflammatory factors and lipid profiles, respectively

FABP2 and zonulin are biomarkers that indicate the status of impaired intestinal permeability [27, 28]. We examined the relationship between these two factors and lipid and inflammatory parameters. Spearman's correlation analysis revealed correlations between FABP2 and endotoxin (r = 0.159, $P = 0.007$), TNF-α (r = 0.313, $P < 0.001$), MCP-1 (r = 0.472, $P < 0.001$), and IP10 (r = 0.356, $P < 0.001$) (Fig 1), as well as hs-CRP (r = 0.116, $P = 0.05$) and HDL (r = −0.128, $P = 0.034$), in the study participants. Zonulin levels were correlated with TG (r = 0.197, $P = 0.001$), HLD (r = −0.181, $P = 0.003$), fasting glucose (r = 0.21, $P < 0.001$), HbA1c (r = 0.322, $P < 0.001$), and hs-CRP (r = 0.858, $P < 0.001$) (Fig 2).

### FABP2 Levels are independently associated with severe hot flashes

Multivariate linear regression model was used to examine the relationship between hot flashes status and gut permeability biomarkers—zonulin and FABP2—after adjusting for BMI, age, and menopause duration. Linear regression analysis revealed that the severity of hot flashes had significant associations with the FABP2 plasma levels ($P$-trend = 0.002), but not with the zonulin levels. Multivariate linear regression analysis further found differences in terms of FABP2 in groups HFs (% difference (95% confidence interval), 22.36 (8.04, 38.59), $P = 0.01$), $HF_M$ (3.16 (−12.62, 21.79), $P > 0.05$), and $HF_m$ (1.60(−14.89, 21.28), $P > 0.05$), compared with

**Table 1. Characteristics of the participants according to hot flashes status.**

| Parameters | Hot flashes status | | | | P-value | P-trend |
|---|---|---|---|---|---|---|
| | HF$_0$ | HF$_m$ | HF$_M$ | HF$_S$ | | |
| n | 117 | 33 | 41 | 98 | — | — |
| Age‡, years | 53.6 ± 4.4 | 54.1 ± 4 | 54 ± 4.9 | 53.9 ± 3.9 | 0.929 | 0.716 |
| MP_duration‡, years | 5.1 ± 4.8 | 3.9 ± 4 | 3.2 ± 3.3 | 4.9 ± 4.9 | 0.102 | 0.538 |
| BMI†, kg/m$^2$ | 23.2 (21.5, 25.6) | 22.6 (21.6, 24.6) | 23.1 (21.2, 26.3) | 24.4 (22.2, 26.7) | 0.063 | 0.043 |
| FSH†, mIU/mL | 57.5 (30, 78) | 67.3 (49.2, 80) | 67.1 (52.5, 81.1) | 62.3 (45, 80.2) | 0.229 | 0.159 |
| Estradiol, pg/mL | <20 | <20 | <20 | <20 | — | — |
| Fasting glucose†, mg/dL | 95 (90, 102) | 95 (90, 100.5) | 98 (91, 108) | 96.5 (90, 103) | 0.805 | 0.715 |
| Hemoglobin A1c†, % | 5.5 (5.3, 5.8) | 5.5 (5.1, 5.8) | 5.5 (5.3, 5.9) | 5.5 (5.4, 5.9) | 0.410 | 0.141 |
| Total cholesterol†, mg/dL | 202 (184, 227) | 200 (185.5, 241) | 202 (170, 226) | 206 (178, 234) | 0.846 | 0.827 |
| Triglyceride†, mg/dL | 92 (68, 127) | 101.5 (69, 125.5) | 96 (65, 110) | 102.5 (73.5, 135) | 0.125 | 0.569 |
| HDL cholesterol†, mg/dL | 57.5 (50, 70) | 55.5 (48.5, 65) | 59 (50, 71) | 55 (47, 63.5) | 0.223 | 0.104 |
| LDL cholesterol†, mg/dL | 122 (103, 145) | 121 (111, 157) | 118 (105, 146) | 122 (107, 150) | 0.667 | 0.721 |
| Zonulin†, ng/ml | 42.6 (8.7, 102.4) | 33.3 (11.2, 99.7) | 47.6 (11.3, 87.1) | 55.8 (15.9, 138.3) | 0.336 | 0.124 |
| FABP2†, ng/ml | 2.2 (1.6, 2.9) | 2.1 (1.7, 2.5) | 2.4 (1.8, 2.7) | 2.6 (1.9, 3.7)* | 0.005 | 0.001 |
| Endotoxin†, EU/ml | 2.6 (1.2, 4.5) | 2.7 (1.4, 4.2) | 3.9 (2.1, 5.5) | 3.4 (1.5, 5.9) | 0.132 | 0.031 |
| hs-CRP†, mg/dL | 0.04 (0.01, 0.17) | 0.02 (0.01, 0.1) | 0.04 (0.01, 0.19) | 0.08 (0.03, 0.27) | 0.018 | 0.033 |
| TNF-α†, pg/ml | 8.8 (6.3, 10.7) | 7.5 (6.2, 12.1) | 8.5 (7.3, 11.3) | 9.3 (7.3, 13.9)* | 0.020 | 0.016 |
| IP10†, pg/ml | 283.5 (217.2, 345) | 270.7 (217.2, 436.1) | 283.7 (206.5, 389.3) | 315.9 (235.8, 424.8) | 0.121 | 0.021 |
| MCP-1†, pg/ml | 143.3 (115.5, 179.4) | 144.2 (121.6, 180) | 141.2 (122.4, 177.1) | 151.7 (123.8, 203.3) | 0.269 | 0.078 |

Data are presented as median (Q1, Q3). Statistical analysis was performed using Kruskal–Wallis test (marked with †) or ANOVA test (marked with ‡) to compare the mean/median differences between the study groups. Dunn's multiple comparison tests and Tukey's post hoc tests were then performed to find significant differences between groups.

*Significant difference between HF$_S$ and HF$_0$ groups ($P < 0.05$).

Abbreviations: HF$_0$: never experienced hot flashes; HF$_m$: mild hot flashes; HF$_M$: moderate hot flashes; HF$_S$: severe hot flashes; Q: quarter; Q1: 25$^{th}$ percentile; Q3: 75$^{th}$ percentile; MP_duration: menopause period since final menstrual period; FSH: follicle-stimulating hormone; BMI: body mass index; HDL: high-density lipoprotein; LDL: low-density lipoprotein; FABP2: fatty acid binding protein 2; TNF-α: tumor necrosis factor alpha; IP10: interferon-inducible protein-10; MCP-1: monocyte chemoattractant protein-1.

group HF$_0$ (Table 2). The associations between the severity of hot flashes and plasma zonulin levels were not significant (Table 2).

## Endotoxin levels are correlated with circulating inflammatory factors

In the present study, we also examined the relationship between circulating endotoxin and TNF-α, MCP1, and IP10. Spearman's correlation analysis revealed that circulating endotoxin levels were highly correlated with TNF-α (r = 0.130, $P = 0.0276$), MCP1 (r = 0.208, $P = 0.0004$), and IP10 (r = 0.101, $P = 0.086$) (Fig 3).

## Discussion

We investigated the relationship between hot flashes and gut permeability and found that hot-flash intensity was positively associated with elevated plasma levels of FABP2 in postmenopausal women. This suggests that severe hot flashes are linked to changes in gut barrier permeability in postmenopausal women.

Over the past decade, the interaction of estrogen with gut microbiota has become a rapidly growing and exciting area of research [24]. Estrogen decline and deficiency observed in

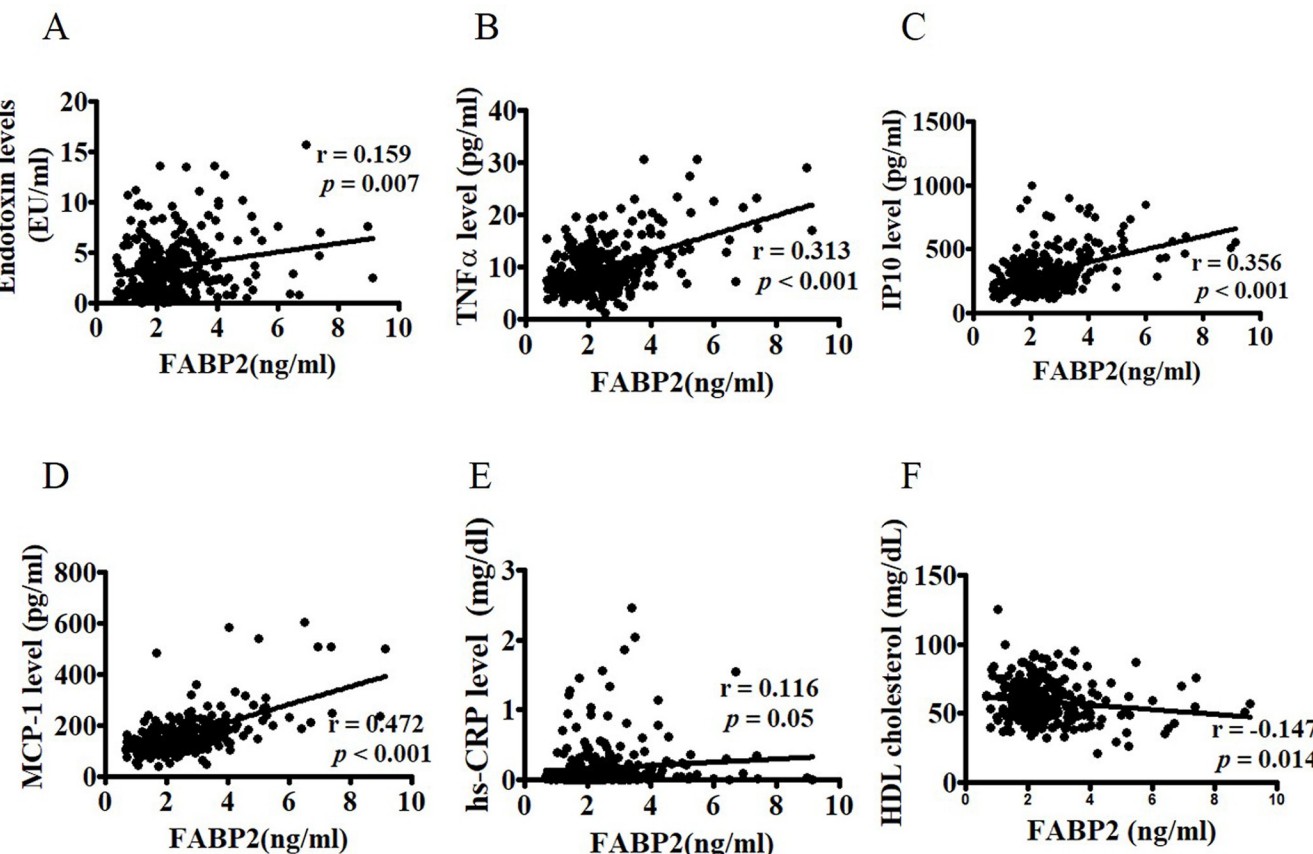

**Fig 1. Correlation between FABP2 and circulating inflammatory factors.** Correlations between FABP2 levels and endotoxin (A), TN-Fα (B), IP10 (C), and MCP-1 (D) as well as hs-CRP € and HDL cholesterol (F) in 289 postmenopausal women were determined using Spearman's correlation analysis. FABP2: fatty acid binding protein 2; TNF-α: tumor necrosis factor alpha; IP10: interferon-inducible protein-10; MCP-1: monocyte chemoattractant protein-1 high-sensitivity C-reactive protein; HDL: high-density lipoprotein.

menopausal women can induce physical functional alterations, such as metabolic and immunological changes through dysbiosis of gut microbiota, characterized by changes in the composition and diversity of intestine microbiome [15, 24]. Gut dysbiosis is associated with increased intestinal permeability [29], which causes the luminal toxic compounds (e.g., lipopolysaccharides) to leak into the systemic circulation. FABP2 and zonulin are two biomarkers of gut epithelium tight junction barrier integrity. In the present study, we found that plasma levels of FABP2, but not zonulin, were higher in women with hot flashes (n = 172) than in women without hot flashes (n = 117)(S1 Fig), suggesting that hot flashes were possibly related to the increased gut permeability specific to FABP2.

Zonulin is a physiological modulator of the intercellular tight junctions in the intestinal tract. It is responsible for the movement of fluid, macromolecules, and leukocytes (e.g., macrophage) between the blood stream and the intestinal lumen [30]. Increased serum levels of zonulin are accompanied with a leaky intestinal barrier, dysbiosis, and inflammation [31] and are observed in many diseases, such as celiac disease [32]. In the present study including in postmenopausal women (n = 289), zonulin levels were significantly correlated with total cholesterol, TG, LDL, and BMI, consistent with previous reports [33] but were unassociated with hot flashes. The possible explanation is that because zonulin is secreted by the intestine and many extraintestinal tissues (e.g., liver and adipose tissues), using commercial ELISA assay, a

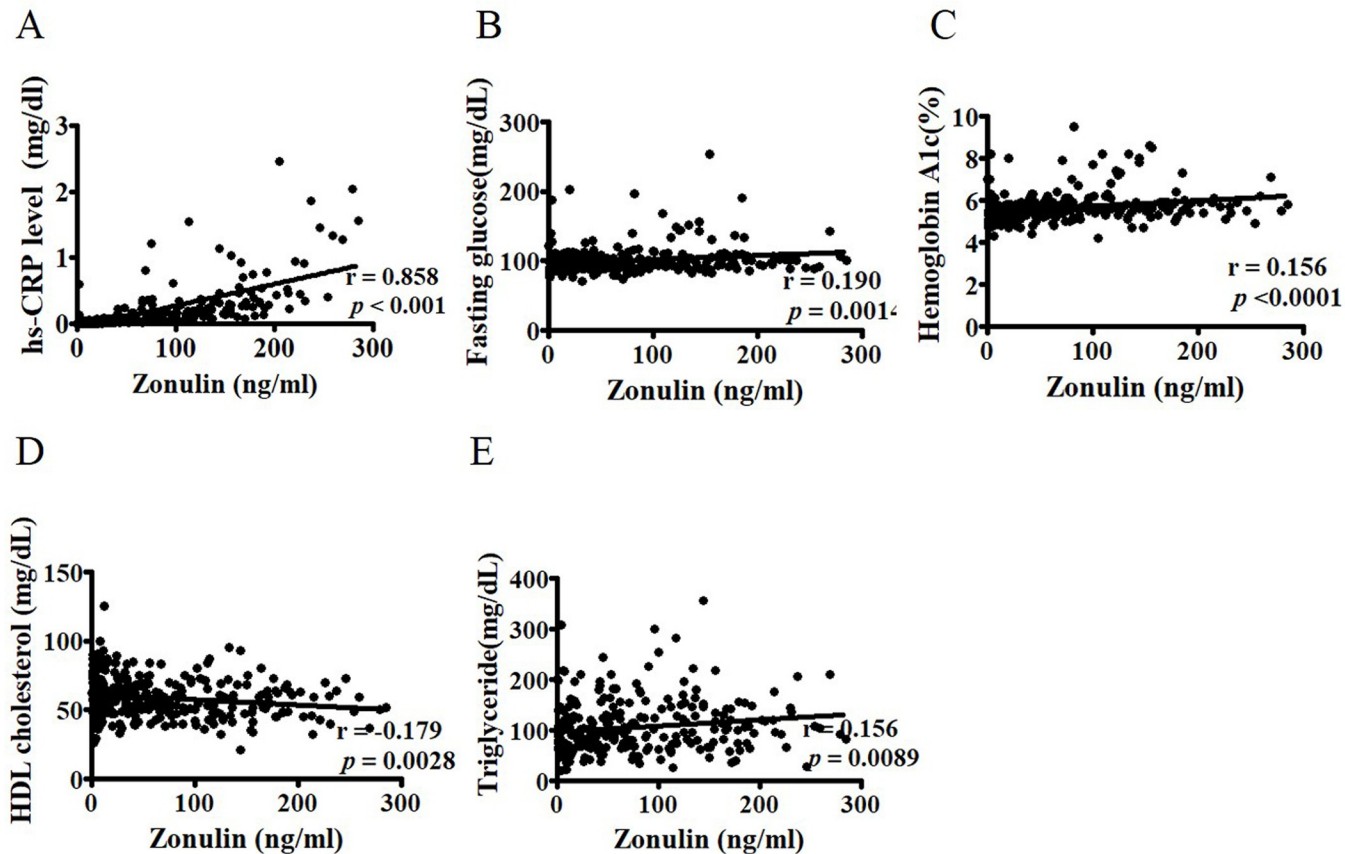

**Fig 2. Correlation between zonulin and hs-CRP, glucose, and lipid profiles.** Correlations between zonulin levels and hs-CRP (A), fasting glucose (B), HbA1c (C), HDL cholesterol (D), and triglycerides (E) in 289 postmenopausal women were determined using Spearman's correlation analysis. HDL: high-density lipoprotein; hs-CRP: high-sensitivity C-reactive protein; HbA1c: glycosylated hemoglobin.

**Table 2. Associations between hot flashes and zonulin and fatty acid binding protein 2.**

| Hot flashes severity | Zonulin | | | | Fatty acid binding protein 2 | | | |
|---|---|---|---|---|---|---|---|---|
| | Unadjusted | | Adjusted | | Unadjusted | | Adjusted | |
| | Coefficient (95% CI) | r | Coefficient (95% CI) | r | Coefficient (95% CI) | r | Coefficient (95% CI) | r |
| $HF_0$ | 1 | 0 | 1 | 0 | 1 | 0 | 1 | 0 |
| $HF_m$ | −7.96 (−50.08, 69.68) | −0.017 | −3.75 (−47.05, 74.97) | −0.008 | 2.97 (−13.64, 22.78) | 0.020 | 1.60 (−14.89, 21.28) | 0.011 |
| $HF_M$ | 16.65 (−33.59, 104.88) | 0.034 | 36.33 (−22.17, 138.79) | 0.068 | 5.51 (−10.27, 24.07) | 0.041 | 3.16 (−12.62, 21.79) | 0.023 |
| $HF_S$ | 43.14 (−6.42, 118.95) | 0.107 | 28.35 (−15.69, 95.39) | 0.075 | 23.76 (9.52, 39.84) | 0.218[c] | 22.36 (8.04, 38.59) | 0.206[b] |
| P-trend | 0.118 | | 0.182 | | 0.004 | | 0.002 | |

Data are expressed as the percentage difference (95% CI).

Regression coefficients are back-transformed using the formula $100^*(\exp(\beta)-1)$ to calculate the percentage difference and the 95% CI in cytokine/chemokine index for hot flashes status per one unit increment.

Linear regression model was adjusted for age, menopause duration, and body mass index.

[a]$P$-value $< 0.05$.

[b]$P$-value $< 0.01$.

[c]$P$-value $< 0.001$

Abbreviations: $HF_0$: never experienced hot flashes; $HF_m$: mild hot flashes; $HF_M$: moderate hot flashes; $HF_S$: severe hot flashes.

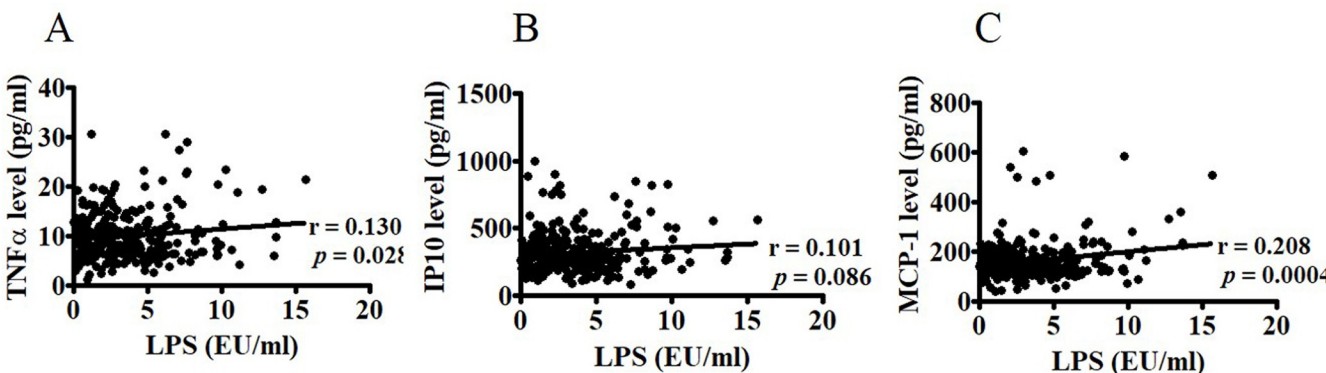

**Fig 3. Associations between circulating endotoxin and inflammatory factors.** Associations between circulating endotoxin levels and TNF-α (A), IP10 B), and MCP-1(C) in 289 postmenopausal women were determined using Spearman's correlation analysis. TNF-α: tumor necrosis factor alpha; IP10: interferon-inducible protein-10; MCP-1: monocyte chemoattractant protein-1.

variety of proteins structurally related to zonulin were identified rather than a single member of intestinal-permeability-regulating proteins [33]. The assay then cannot represent a culmination of zonulin in the intestinal mucosa, affected by the hot flashes. Further studies are necessary to establish assays that specifically target zonulin produced from the intestinal barrier as a marker for intestinal permeability.

FABP2 is a form of 14–15 kD cytoplasmic protein and is involved in the intracellular metabolism and transport of long-chain fatty acids. FABP2 is also known as intestinal-type fatty acid binding protein (I-FABP). Several studies have described the use of FABP2 in monitoring intestinal injuries [34]. Both experimental and clinical studies demonstrated that FABP2 is a surrogate biomarker of intestinal barrier function [35–37]. In the present study, we found that FABP2 levels were correlated with circulating endotoxin levels and inflammatory factors, including TNFα, MCP-1, and IP10, in postmenopausal women (Fig 2). Furthermore, TNFα, MCP-1, and IP10 were significantly correlated with endotoxin (Fig 3), suggesting that FABP2 is a marker of increased gut permeability in inflammation-related leaks in postmenopausal women [27].

Hot flashes are the most common symptom of menopausal syndrome. Gut microbiota undeniably play a crucial role in maintaining intestinal physical barrier function and preventing disease progression [38]. The imbalance of the beneficial and detrimental microorganisms can cause dysbiosis. Although dysbiosis is well identified in menopausal women [39], few studies reported the association of the intestinal barrier permeability with hot flashes. A number of characteristics of the hot flashes, including severity and frequency, may contribute to health outcomes such as cardiovascular and metabolic disorders in menopausal women [9, 11]. In this study, we report for the first time that severe hot flashes are an independent factor associated with the integrity of the gastrointestinal barrier in postmenopausal women (Table 2).

Our previous and other studies have demonstrated that hot flashes are linked to metabolic disorders, such as insulin resistance and adiponectin/leptin imbalance [9] and low-grade systemic inflammation [26]. The potential mechanisms underlying the association between metabolic disorders and hot flashes remain unclear. Recently, intestinal microbiota is recognized as a key environmental factor in metabolic diseases [40]. Indeed, the gut microbiota is considered a separate endocrine organ, involved in the metabolic and immune homeostasis, through a molecular crosstalk with the host [41]. We detected increased levels of proinflammatory factors and endotoxin in participants with hot flashes, which was positively correlated with FABP2. This result may support the hypothesis that the presented associations between

systemic inflammation and hot flashes may stem from increased gut permeability. Thereby, the increased low-grade systemic inflammation could promote the development of insulin resistance and metabolic disorders [42]. However, the mechanisms explaining the association between the variations in the composition and diversity of the gut microbiome and the development of the metabolic inflammation in women who experience hot flashes remain elusive. Further studies are needed to identify the complex etiology of such pathologies.

Several limitations exist in this study, which need to be addressed. First, since the present study is cross-sectional, the measurement was insufficient to evaluate if a causal relationship exists between gut permeability and hot flashes. Second, zonulin is an established serum marker for intestinal permeability. However, zonulin levels were measured using a widely used commercial ELISA kit that may recognize structural analog proteins but not prehaptoglobin 2 [33]. Third, several contributing factors are most likely involved in the increased gut permeability, including chronic constipation, chronic bowel disorders, as well as hepatic and biliary tract diseases [43–46]. There was no obvious medical history of acute and chronic bowel disorders, high-degraded systemic inflammation and bacterial infections in our subjects. However, the constipation and diarrhea symptoms in subjects could not be detected in detail, which may make bias in the present results. Therefore, the results need to be interpreted with such findings in consideration.

In conclusion, we found that hot flashes were significantly associated with elevated levels of circulating FABP2 in postmenopausal women. Such a result evidences the association between hot flashes and gut permeability and low-grade systemic inflammation. Further longitudinal studies are required to clarify the causal relationships between dybiosis, gut permeability, and related inflammation and hot flashes in menopausal women.

## Supporting information

**S1 Fig. Plasma levels of zonulin and FABP2 in menopausal women with/without hot flashes.** The plasma levels of gut barrier protein FABP2, but not zonulin are higher in postmenopausal women with hot flashes (n = 172) than those in women without hot flashes (n = 117) (Student's t-test; $P$ = 0.014).
(TIF)

## Acknowledgments

The authors thank all of the participants in this study.

## Author Contributions

**Conceptualization:** Hung-Ming Wu.

**Data curation:** Wan-Yu Huang, Chew-Teng Kor.

**Formal analysis:** Chew-Teng Kor, Hung-Ming Wu.

**Funding acquisition:** Hung-Ming Wu.

**Investigation:** Ting-Yu Chen, Ko-Hung Liu, Yi-Chun Chao.

**Project administration:** Ting-Yu Chen.

**Supervision:** Hung-Ming Wu.

**Writing – original draft:** Wan-Yu Huang.

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
