## [Decision Letter · Decision Letter 0]

6 Apr 2022

PONE-D-21-34256

The Relationship between Hot Flashes and Fatty Acid Binding Protein 2 in Postmenopausal Women

PLOS ONE

Dear Dr. Wu,

Thank you for submitting your manuscript to PLOS ONE. After careful consideration, we feel that it has merit but does not fully meet PLOS ONE’s publication criteria as it currently stands. Therefore, we invite you to submit a revised version of the manuscript that addresses the points raised during the review process.

We look forward to receiving your revised manuscript.

Kind regards,

Jing Zhang

Academic Editor

PLOS ONE

https://journals.plos.org/plosone/s/file?id=ba62/PLOSOne_formatting_sample_title_authors_affiliations.pdf".

2. We noted in your submission details that a portion of your manuscript may have been presented or published elsewhere. [DETAILS AS NEEDED] Please clarify whether this [conference proceeding or publication] was peer-reviewed and formally published. If this work was previously peer-reviewed and published, in the cover letter please provide the reason that this work does not constitute dual publication and should be included in the current manuscript.

Reviewers' comments:

Reviewer's Responses to Questions

**Comments to the Author**

1. Is the manuscript technically sound, and do the data support the conclusions?

Reviewer #1: Yes

Reviewer #2: Yes

2. Has the statistical analysis been performed appropriately and rigorously? 

Reviewer #1: Yes

Reviewer #2: Yes

3. Have the authors made all data underlying the findings in their manuscript fully available?

Reviewer #1: Yes

Reviewer #2: Yes

4. Is the manuscript presented in an intelligible fashion and written in standard English?

Reviewer #1: Yes

Reviewer #2: No

5. Review Comments to the Author

Reviewer #1: 1. The results of the study evidence the association between hot flashes and gut permeability and low-grade systemic inflammation. The authors also propose the possible causal relationships between dysbiosis, gut permeability, related inflammation and hot flashes in menopausal women. This raises a concern about the study design. Do you exclude the postmenopausal patients with GI diseases, such as diarrhea, constipation, liver diseases, biliary tract diseases, inflammatory bowel diseases? Because these conditions may cause disrupted gut permeability and systemic inflammation. And they make bias in the results. Please explain and discuss it in the revised manuscript.

Reviewer #2: The authors study the relationship between hot flashes (and their severity) and gut permeability. Their study is innovative and well carried out.

I do have a few comments.

In the introduction the authors write: To gain a better understanding of the relationship between hot flashes and

gut-microbiota-derived mechanisms, a case-control study was conducted in age-matched groups

of postmenopausal women with and without hot flashes.

a - this study is cross sectional and not a case control

b - I suggest to more clearly define the objective (as defined in the discussion section) to investigate the relationship between hot flashes and gut permeability

Materials and methods: It is not clear at what time period where the women recruited (years and months). It is also not clear if all postmenopausal women who attended the hospital were offered participation? Visitors? Patients? Women attending a specific clinic?

The authors write: Linear regression analysis revealed that the severity of hot flashes had significant effects on FABP2

I recommend writing association as no causality and cause effect can be concluded

Discussion

The authors write: "In the present study, we found that plasma levels of zonulin and FABP2 (biomarkers of gut permeability) were higher in women with hot flashes (n = 172) than in women without hot flashes (n = 117)(S1 Fig), suggesting that hot flashes were possibly related to gut permeability changes".

However, no association was found with zonulin levels .

Also - the association between FAB2 and hot flashes was found groups HFs whereas in HFM and HFm the association was not significant (p>0.05)

Please clarify the term metacardiovascular diseases

The authors write: Although dysbiosis is well identified in menopausal women, few studies reported its association with hot flashes. A number of characteristics of the hot flashes, including severity and frequency, may contribute to health outcomes such as metacardiovascular diseases in menopausal women [9, 11]. In this study, we report for the first time that severe hot flashes are an independent factor associated with the integrity of the gastrointestinal barrier in postmenopausal women (Table 2).

I am not sure if this paragraph is not clear grammatically or for some other reason - but this study too (as far as I understood) did not check for an association between dysbiosis/ gut microbiota and hot flashes. Rather it checked the association with gut permeability.

A few typos and grammatical errors are present - please spell check and also check for grammar

6. PLOS authors have the option to publish the peer review history of their article (what does this mean?). If published, this will include your full peer review and any attached files.

Reviewer #1: **Yes: **Dr. Tsung-Hsuan Lai, M.D., Ph.D.

Reviewer #2: No

---

## [Author Response · Author response to Decision Letter 0]

17 May 2022

Response to reviewers’ comments:

We would like to thank the academic editor and reviewers for the comprehensive assessment of our manuscript, giving very constructive criticism and comments. We have taken all the remarks into account, in a manner that is described in detail below together with our response to certain comments. We think that, following the reviewers’ suggestions, our manuscript has gained in clarity and hope that the changes made will be considered satisfactory.

Reviewers' comments:

Reviewer's Responses to Questions

Comments to the Author

1. Is the manuscript technically sound, and do the data support the conclusions?

Reviewer #1: Yes

Reviewer #2: Yes

 2. Has the statistical analysis been performed appropriately and rigorously? 

 Reviewer #1: Yes

Reviewer #2: Yes

 3. Have the authors made all data underlying the findings in their manuscript fully available?

 Reviewer #1: Yes

Reviewer #2: Yes

 4. Is the manuscript presented in an intelligible fashion and written in standard English?

 Reviewer #1: Yes

Reviewer #2: No

Response:

1. We thank you very much for useful suggestion.

2. The manuscript has been revised by a native English speaker with extensive experience in medical and scientific editing.________________________________________

 5. Review Comments to the Author

Reviewer #1: 1. The results of the study evidence the association between hot flashes and gut permeability and low-grade systemic inflammation. The authors also propose the possible causal relationships between dysbiosis, gut permeability, related inflammation and hot flashes in menopausal women. This raises a concern about the study design. Do you exclude the postmenopausal patients with GI diseases, such as diarrhea, constipation, liver diseases, biliary tract diseases, inflammatory bowel diseases? Because these conditions may cause disrupted gut permeability and systemic inflammation. And they make bias in the results. Please explain and discuss it in the revised manuscript.

Response

1. Thank the reviewer very much for the critical comments. 

2. The mechanisms of action for leaky gut are still not fully determined. Multiple contributing factors are most likely involved in the increased gut permeability, including chronic constipation, chronic bowel disorders, and hepatic biliary tract diseases (Fukui, Inflamm Intest Dis. 2016;1:135-145; Vanuytsel, Front Nutr. 2021;8:717925; Hanning, Therap Adv Gastroenterol. 2021;14:1756284821993586; Ohkusa, Front Med. 2019;6:19) as the Reviewer’s mention.

3. We have carefully reviewed the medical history of our cases based on the reviewer’s comments. Among those subjects in the present study, there was no obvious medical history of acute and chronic bowel disorders. Their hs-CRP values were less than 10 mg/dl (Table 1), indicating no high-degraded systemic inflammation and bacterial infections. However, the constipation and diarrhea symptoms in these subjects could not be detected in detail, which may make bias in the present results. 

4. According to your nice suggestions, several sentences representing the limitations have been mentioned and added into the limitation paragraph (Please see Page 16).

Reviewer #2: The authors study the relationship between hot flashes (and their severity) and gut permeability. Their study is innovative and well carried out.

I do have a few comments.

In the introduction the authors write: To gain a better understanding of the relationship between hot flashes and

gut-microbiota-derived mechanisms, a case-control study was conducted in age-matched groups

of postmenopausal women with and without hot flashes.

a - this study is cross sectional and not a case control

b - I suggest to more clearly define the objective (as defined in the discussion section) to investigate the relationship between hot flashes and gut permeability

Response

1. Thank the reviewer very much for very variable suggestions. 

2. These points have been modified in the Introduction section (Please see Page 6).

Materials and methods: It is not clear at what time period where the women recruited (years and months). It is also not clear if all postmenopausal women who attended the hospital were offered participation? Visitors? Patients? Women attending a specific clinic?

The authors write: Linear regression analysis revealed that the severity of hot flashes had significant effects on FABP2

I recommend writing association as no causality and cause effect can be concluded

Response

1. Thank the reviewer very much for very useful suggestions. 

2. We have added the information of the participants, who visited for health management reasons in the section of Materials and Methods (Please see Page 7).

3. In addition, the sentence “Linear regression analysis revealed that the severity of hot flashes had significant effects on FABP2” has been modified as “Linear regression analysis revealed that the severity of hot flashes had significant associations with the FABP2 plasma levels (P-trend = 0.002), but not with the zonulin levels” in the Result section (Please see Page 12).

Discussion

The authors write: "In the present study, we found that plasma levels of zonulin and FABP2 (biomarkers of gut permeability) were higher in women with hot flashes (n = 172) than in women without hot flashes (n = 117)(S1 Fig), suggesting that hot flashes were possibly related to gut permeability changes".

However, no association was found with zonulin levels.

Also - the association between FAB2 and hot flashes was found groups HFs whereas in HFM and HFm the association was not significant (p>0.05)

Response

1. Thank the reviewer very much for very variable comments. 

2. There is a mistake in this sentence “In the present study, we found that plasma levels of zonulin and FABP2 (biomarkers of gut permeability) were higher in women with hot flashes (n = 172) than in women without hot flashes (n = 117)(S1 Fig), suggesting that hot flashes were possibly related to gut permeability changes". It has been correlated as “--- plasma levels of FABP2, but not zonulin, were higher in women with hot flashes (n = 172) than in women without hot flashes (n = 117)(S1 Fig),---.” We also modified several sentences in this paragraph (Please see this paragraph in the Discussion section, Page 13)

3. Linear regression analysis found that FABP2 levels had positive tendency to the severity of hot flashes, and multiple linear regression analysis further revealed the significant relationships between “severe” hot-flash (HFs) group and FABP2 levels. According to your nice suggestions, several sentences have been modified in the paragraph of “FABP2 Levels are independently associated with hot flashes” (Please see this paragraph in the Result section, Page 12).

Please clarify the term metacardiovascular diseases

I am not sure if this paragraph is not clear grammatically or for some other reason - but this study too (as far as I understood) did not check for an association between dysbiosis/ gut microbiota and hot flashes. Rather it checked the association with gut permeability.

Response

1. Thank the reviewer very much for very useful suggestions. 

2. This mistake has been modified as “cardiovascular and metabolic disorders” (Please see Page 15), which includes cardiovascular diseases and metabolic diseases, such as type 2 diabetes.

3. We agree with the Reviewer’s comment. The present study did not check for an association between dysbiosis/ gut microbiota and hot flashes. Therefore, this paragraph has been modified (Please see this paragraph in the Discussion section, Pages 14 and 15)

A few typos and grammatical errors are present - please spell check and also check for grammar

Response:

1. We thank you very much for useful suggestion. 

2. The manuscript has been revised by a native English speaker with extensive experience in medical and scientific editing.________________________________________

 6. PLOS authors have the option to publish the peer review history of their article (what does this mean?). If published, this will include your full peer review and any attached files.

Do you want your identity to be public for this peer review? For information about this choice, including consent withdrawal, please see our Privacy Policy.

 Reviewer #1: Yes: Dr. Tsung-Hsuan Lai, M.D., Ph.D.

Reviewer #2: No

---

## [Decision Letter · Decision Letter 1]

13 Jul 2022

PONE-D-21-34256R1The Relationship between Hot Flashes and Fatty Acid Binding Protein 2 in Postmenopausal WomenPLOS ONE

Dear Dr. Wu,

Thank you for submitting your manuscript to PLOS ONE. After careful consideration, we feel that it has merit but does not fully meet PLOS ONE’s publication criteria as it currently stands. Therefore, we invite you to submit a revised version of the manuscript that addresses the points raised during the review process.

We look forward to receiving your revised manuscript.

Kind regards,

Jing Zhang

Academic Editor

PLOS ONE

**Comments to the Author**

1. If the authors have adequately addressed your comments raised in a previous round of review and you feel that this manuscript is now acceptable for publication, you may indicate that here to bypass the “Comments to the Author” section, enter your conflict of interest statement in the “Confidential to Editor” section, and submit your "Accept" recommendation.

Reviewer #2: All comments have been addressed

Reviewer #3: (No Response)

2. Is the manuscript technically sound, and do the data support the conclusions?

Reviewer #2: Yes

Reviewer #3: Yes

3. Has the statistical analysis been performed appropriately and rigorously? 

Reviewer #2: Yes

Reviewer #3: Yes

4. Have the authors made all data underlying the findings in their manuscript fully available?

Reviewer #2: Yes

Reviewer #3: Yes

5. Is the manuscript presented in an intelligible fashion and written in standard English?

Reviewer #2: Yes

Reviewer #3: Yes

6. Review Comments to the Author

Reviewer #2: (No Response)

Reviewer #3: The authors showed that there is an association with hot flashes and gut permeability/inflammation.

The major concern with this paper is the measurement of endotoxin. Endotoxin is very difficult to measure because there is contamination of it everywhere, therefore it is almost imposible to measure endotoxin unless the samples are collected in an endotoxin free environment and are measured in an endotoxin free facility. Therefore, with the data presented in this paper it will be hard to conclude that there is increased gut permeability just by measuring FABP2. There are other ways to measure presence of microbial translocation such as LBP (LPS binding protein) or sCD14 (receptor of LPS). Therefore the conclusions made by the authors are weakened by this flaw. The conclusions will be strengthened if other markers of gut permeability/microbial translocation in addition of FABP2 are measured. Nonetheless, it is interesting that inflammation is also associated with hot flashes however the connexion of inflammation with gut permeability is weak.

7. PLOS authors have the option to publish the peer review history of their article (what does this mean?). If published, this will include your full peer review and any attached files.

Reviewer #2: No

Reviewer #3: No

---

## [Author Response · Author response to Decision Letter 1]

5 Aug 2022

PONE-D-21-34256R1

The Relationship between Hot Flashes and Fatty Acid Binding Protein 2 in Postmenopausal Women

Response to reviewers’ comments:

We would like to thank the academic editor and reviewers for the comprehensive assessment of our manuscript, giving very constructive criticism and comments. We have taken all the remarks into account, in a manner that is described in detail below together with our response to certain comments. We think that, following the reviewers’ suggestions, our manuscript has gained in clarity and hope that the changes made will be considered satisfactory.

Reviewers' comments:

Reviewer's Responses to Questions

Comments to the Author

Reviewer #2: (No Response)

Reviewer #3: The authors showed that there is an association with hot flashes and gut permeability/inflammation.

The major concern with this paper is the measurement of endotoxin. Endotoxin is very difficult to measure because there is contamination of it everywhere, therefore it is almost imposible to measure endotoxin unless the samples are collected in an endotoxin free environment and are measured in an endotoxin free facility. Therefore, with the data presented in this paper it will be hard to conclude that there is increased gut permeability just by measuring FABP2. There are other ways to measure presence of microbial translocation such as LBP (LPS binding protein) or sCD14 (receptor of LPS). Therefore the conclusions made by the authors are weakened by this flaw. The conclusions will be strengthened if other markers of gut permeability/microbial translocation in addition of FABP2 are measured. Nonetheless, it is interesting that inflammation is also associated with hot flashes however the connexion of inflammation with gut permeability is weak.

Response:

1. Thank the reviewer very much for the critical comments and suggestions. 

2. We agree with the concerns as the reviewer mentioned. Endotoxin is very difficult to measure because there is contamination of it everywhere, leading to unreliable experimental results in diagnostic and research laboratories.

3. In our Lab, we have a lot of experience in endotoxin assay and endotoxin experiments (Wu CL et al., PLoS One. 2014 Oct 6;9(10):e109558; Chang CC et al., Sci Rep. 2015;5:10096; Shih KL et al., Sci Rep. 2016;6:24031; Huang WY et al., J Neuroinflammation. 2022 Feb 2;19(1):29).

4. In our Lab, Chromogenic Limulus Amebocyte Lysate Assay (QCL-1000™; Lonza, Walkersville, MD, USA) was used for endotoxin detection. From sample collection to endotoxin detection, each procedure was performed in an endotoxin free facility, by using endotoxin-free materials and reagents, including nonpyrogenic plasticware, depyrogenated glassware, and high-purity water. These steps will prevent the most common sources of endotoxin contamination and ensure the reliability of the obtained results. Therefore, several sentences representing the concern for endotoxin contamination been added into the “Measurements of circulating endotoxin levels” paragraph (Please see Page 9).

5. LBP and sCD14 are two proteins that promote innate immunity to Gram-negative bacteria by transferring LPS to TLR4. Although LBP is synthesized by intestinal cells as well as the liver and adipose tissue, and sCD14 is secreted by the liver and monocytes, both are the potential biomarkers that could be correlated with LPS levels, and related to intestinal bacteria exposure. As the reviewer’s suggestion, LPS in blood is an important and feasible biomarker of intestinal barrier dysfunction. If other microbial translocation markers (e.g., LBP) are measured in the blood, it may provide stronger information for gut permeability. Agreeably, microbial translocation markers and sCD14 are considered to be measured for future related studies.

---

## [Decision Letter · Decision Letter 2]

6 Oct 2022

The Relationship between Hot Flashes and Fatty Acid Binding Protein 2 in Postmenopausal Women

PONE-D-21-34256R2

Dear Dr. Wu,

We’re pleased to inform you that your manuscript has been judged scientifically suitable for publication and will be formally accepted for publication once it meets all outstanding technical requirements.

Kind regards,

Jing Zhang

Academic Editor

PLOS ONE

Additional Editor Comments (optional):

Reviewers' comments:

Reviewer's Responses to Questions

**Comments to the Author**

1. If the authors have adequately addressed your comments raised in a previous round of review and you feel that this manuscript is now acceptable for publication, you may indicate that here to bypass the “Comments to the Author” section, enter your conflict of interest statement in the “Confidential to Editor” section, and submit your "Accept" recommendation.

Reviewer #3: All comments have been addressed

2. Is the manuscript technically sound, and do the data support the conclusions?

Reviewer #3: Yes

3. Has the statistical analysis been performed appropriately and rigorously? 

Reviewer #3: Yes

4. Have the authors made all data underlying the findings in their manuscript fully available?

Reviewer #3: Yes

5. Is the manuscript presented in an intelligible fashion and written in standard English?

Reviewer #3: Yes

6. Review Comments to the Author

Reviewer #3: The author has responded to the concerns. It would have been optimal to measure other markers of microbial translocation to make the conclusions more sound but the answer is sufficient.

7. PLOS authors have the option to publish the peer review history of their article (what does this mean?). If published, this will include your full peer review and any attached files.

Reviewer #3: No

---

## [Editor Report · Acceptance letter]

10 Oct 2022

PONE-D-21-34256R2 

The Relationship between Hot Flashes and Fatty Acid Binding Protein 2 in Postmenopausal Women 

Dear Dr. Wu:

I'm pleased to inform you that your manuscript has been deemed suitable for publication in PLOS ONE. Congratulations! Your manuscript is now with our production department. 

Kind regards, 

on behalf of

Dr. Jing Zhang 

Academic Editor

PLOS ONE